# USP14 Regulates DNA Damage Response and Is a Target for Radiosensitization in Non-Small Cell Lung Cancer

**DOI:** 10.3390/ijms21176383

**Published:** 2020-09-02

**Authors:** Arishya Sharma, Alexandru Almasan

**Affiliations:** 1Department of Cancer Biology, Lerner Research Institute, Cleveland Clinic, Cleveland, OH 44195, USA; 2Department of Radiation Oncology, Taussig Cancer Institute, Cleveland Clinic, Cleveland, OH 44195, USA; 3Case Comprehensive Cancer Center, Case Western Reserve University School of Medicine, Cleveland, OH 44106, USA

**Keywords:** USP14, radiosensitization, NSCLC, non-homologous end-joining, homologous recombination

## Abstract

Non-small cell lung cancer (NSCLC) represents ~85% of the lung cancer cases. Despite recent advances in NSCLC treatment, the five-year survival rate is still around 23%. Radiotherapy is indicated in the treatment of both early and advanced stage NSCLC; however, treatment response in patients is heterogeneous. Thus, identification of new and more effective treatment combinations is warranted. We have identified Ubiquitin-specific protease 14 (USP14) s a regulator of major double-strand break (DSB) repair pathways in response to ionizing radiation (IR) by its impact on both non-homologous end joining (NHEJ) and homologous recombination (HR) in NSCLC. USP14 is a proteasomal deubiquitinase. IR treatment increases levels and DSB recruitment of USP14 in NSCLC cell lines. Genetic knockdown, using shUSP14 expression or pharmacological inhibition of USP14, using IU1, increases radiosensitization in NSCLC cell lines, as determined by a clonogenic survival assay. Moreover, shUSP14-expressing NSCLC cells show increased NHEJ efficiency, as indicated by chromatin recruitment of key NHEJ proteins, NHEJ reporter assay, and increased IR-induced foci formation by 53BP1 and pS2056-DNA-PKcs. Conversely, shUSP14-expressing NSCLC cells show decreased RPA32 and BRCA1 foci formation, suggesting HR-deficiency. These findings identify USP14 as an important determinant of DSB repair in response to radiotherapy and a promising target for NSCLC radiosensitization.

## 1. Introduction

Lung cancer is one of the most common malignancies worldwide and has become the leading cause of cancer-related death in USA. Approximately 80–85% cases of lung cancer are diagnosed as non-small cell lung cancer (NSCLC). Radiation therapy (RT) is an effective therapeutic modality for NSCLC for many patients and is indicated in the treatment of both early and advanced stage disease [1,2]. However, NSCLC response to RT is heterogeneous, even in tumors with similar clinical and histological features [3].

Unrepaired DNA double-strand breaks (DSB)s are responsible for the majority of cell death in response to ionizing radiation (IR). Thus, the molecular mechanisms regulating DNA damage response (DDR) pathways are the focus of intensive study as they are likely to yield novel therapeutic combination strategies. There are two main pathways for DNA DSB repair—homologous recombination (HR) and non-homologous end-joining (NHEJ). While activation of HR in the G1 phase of the cell cycle would result in genomic instability arising from chromosomal translocations or gene inversions, unscheduled NHEJ would be mutagenic due to its inherent error-prone nature or its detrimental effect on the exposed telomeres during the mitotic phase. Thus, an appropriate pathway choice is tightly regulated throughout the cell cycle of both normal and cancer cells to maintain cellular genomic stability [4,5,6].

The success in targeting HR deficiencies in cancers is an attractive strategy for identifying new treatment targets. Expression of many HR-associated genes has been correlated with the response to radiotherapy, e.g., elevated expression of RAD51, NBS, and XRCC3 confers radioresistance, while reduced XRCC2 expression confers radiosensitivity [7,8]. Furthermore, genetic polymorphisms leading to HR deficiency, such as single nucleotide polymorphisms in RAD51 and XRCC2, have been shown to be prognostic factors for overall survival in response to IR in NSCLC [7,8]. However, a major limitation is the modest number of cancers that are intrinsically HR deficient. In a recent study, BRCA1/2 mutations were reported to occur in ~2.1% of patients with advanced NSCLC [9]. A pan-cancer analysis of over 100 HR core and related genes revealed that only ~5% of NSCLC cases harbor bi-allelic alterations in a known HR target gene [10]. Pharmacological inhibition of an HR-promoting druggable target, therefore, has the potential to address the majority of NSCLC cases that are HR proficient, and may expand the benefits of RT.

Ubiquitination signaling at DSB sites by the E3 ubiquitin ligases RNF8 and RNF168 is crucial in determining DNA repair pathway choice by recruiting p53-binding protein 1 (53BP1) to DSBs. 53BP1, together with its partner protein(s) e.g., Rap1-interacting factor 1 (RIF1), inhibits BRCA1 (Breast Cancer gene 1)-CtIP (CTBP interacting protein) complex-dependent DSB end resection. This promotes rapid NHEJ of the DSB ends and inhibits the HR pathway. On the other hand, the function of 53BP1 is suppressed by BRCA1 in the S/G2 phase of the cell cycle to allow the more accurate HR pathway to repair DSBs [4,5,6]. By acting as ubiquitin isopeptidases, deubiquitinating (DUB) enzymes play an important role in the regulation of DDR, and are, therefore, potential targets to increase radiosensitization [11].

USP14 is one of the three proteasome-associated deubiquitinase (DUB) enzymes, USP14 and UCHL5 being the two that are associated reversibly with the proteasome [12]. By releasing ubiquitin from its substrates, USP14 promotes free ubiquitin recycling and acts as a check against proteasome degradation of ubiquitin [10,11,12,13]. In addition, USP14 acts as an allosteric regulator of proteasome, independent of its catalytic activity, and has been shown to both activate and inhibit substrate degradation [13]. Depletion of USP14 is known to both up- and down-regulate levels of the substrate protein, and decreases available free ubiquitin pools [14,15,16]. USP14 has a prominent role in NSCLC and several other cancers [17,18]. The pharmacological inhibitor of USP14, VLV1570, is already in clinical development for multiple myeloma, and it is effective in ovarian cancer [19,20]. Recently, we have identified a novel nuclear role of USP14 as a negative regulator of NHEJ and DDR-associated ubiquitination signaling in autophagy-deficient prostate cancer (PCa) cells [21,22]. 

Whether USP14 is a molecular target that enhances radiosensitization in NSCLC is presently unclear. Moreover, nothing is known about the role of USP14 in regulating HR. Since RT is an important treatment modality in NSCLC, this study was undertaken with a goal to determine the possible roles of USP14 in DDR in NSCLC, and to establish whether it is a potential target to increase radiosensitization in NSCLC. This study identifies USP14 as a critical DDR factor in NSCLC. We show that USP14 is recruited to DNA DSBs in response to IR and that it regulates both NHEJ and HR repair. Importantly, targeting USP14 is a potential mechanism to enhance radiosensitization in NSCLC.

## 2. Results

### 2.1. High USP14 Expression Correlates with Poor Prognosis in NSCLC

We first asked whether USP14 is a relevant target in NSCLC using publicly available databases. We first searched the Gene Expression Profiling Interactive Analysis (GEPIA) database to identify the expression pattern of USP14 in normal lung vs. NSCLC. We found that USP14 was significantly (*p*-value cut-off 0.01) up-regulated in both lung adenocarcinoma (LUAD) and lung squamous cell carcinoma (LUSC) compared to normal lung tissue (Figure 1A) [23]. Furthermore, we analyzed the RNA levels of USP14 in 517 LUAD and 502 LUSC tumor samples using the transcriptome expression data from lung cancer patients in TCGA. We found the expression of USP14 levels to be high in both LUAD (8%) and LUSC (10%) (Figure 1B). Moreover, analysis of patient survival data in the KM plotter database shows that high USP14 expression leads to reduced overall survival (OS) in NSCLC patients (Figure 1C). These data are consistent with those of previously published studies of USP14 expression in NSCLC [17] and confirm that USP14 is a potential oncogenic target in NSCLC.

### 2.2. IR Increases Protein Expression and Double-Strand Break Recruitment of USP14

Next, to study the potential role of USP14 in regulating DSB repair in NSCLC cells, we first investigated whether USP14 is recruited to DSBs and whether IR treatment has any effect on USP14 protein expression. Immunostaining and confocal microscopy was used to investigate IR-induced foci (IRIF) formation by USP14. IR treatment, indeed, induced USP14 foci formation in a time-dependent manner in H460 and A549 NSCLC cells, respectively (Figure 2A,B), similar to what we have previously shown in PCa cells [21,22].

Next, immunoblotting analyses show that IR treatment induces up-regulation of USP14 protein levels in both H460 (Figure 2C) and A549 (Figure 2D) NSCLC cells. It may be noted that while we did not detect any appreciable changes in response to IR in the protein levels of UCHL5, another proteasomal associated DUB [12] in H460 cells (Figure 2C), A549 cells had no detectable levels of UCHL5 (Figure 2D).

Next, to confirm whether the nuclear foci formed by USP14 in response to IR are DSB associated we did a co-localization analysis of USP14 with γH2AX, which is a gold-standard marker of DSBs [24,25], using co-immunostaining and confocal microscopic analysis. To demonstrate the generality of the scope of this study we also extended our findings to two other NSCLC cell lines, EBC1, an LUSC cell line, and PC9, an LUAD cell line with mutant EGFR. As shown in Figure 2E, in all four NSCLC cell lines that we tested at 1 h following 5 Gy IR treatment, USP14 foci were formed, which did co-localize with γH2AX foci. Overall, consistent with our previous findings in PCa, these data establish USP14 as a DSB-associated factor in NSCLC.

### 2.3. Targeting USP14 Causes Radiosensitization of NSCLC Cells

Given its role as a DDR factor, we next examined the effect of targeting USP14 on IR-induced cell death in NSCLC. By genetic depletion using short hairpin RNA (shRNA)-mediated knock down of USP14, the clonogenic survival in response to IR treatment was significantly reduced in H460 (Figure 3A, *p* < 0.05) and A549 (Figure 3B, *p* < 0.05) cells.

To determine whether the catalytic activity of USP14 is needed for the observed effect on radiosensitization, clonogenic survival analysis was performed in response to pharmacological inhibition of USP14 by IU1 pretreatment prior to IR in these NSCLC cell lines. IU1 treatment in the absence of IR induced varying degrees of decrease in clonogenic survival in various NSCLC cell lines, i.e., ~50% in H460 (Figure 3C) and A549 (Figure 3D), no effect in PC9 (Figure 3E), and >90% in EBC1 (Figure 3F). Nevertheless, inhibition of USP14 resulted invariably in significantly increased radiosensitization in all the NSCLC cell lines tested (Figure 3C–E, *p*-values < 0.01, 0.01, 0.001, respectively) except in EBC1, where significance could not be conclusively established due to a high percentage of cell death with IU1 treatment alone. Taken together, these data show that USP14 is a potential druggable target to increase radiosensitization in NSCLC.

### 2.4. USP14 Disrupts NHEJ

We found that targeting USP14 leads to radiosensitization in NSCLC cells, suggesting an inefficient repair of DSBs in the absence of USP14. Therefore, we investigated the role of USP14 regulating NHEJ in these NSCLC cell lines.

First, biochemical fractionation was used to study chromatin recruitment of RNF168 and other NHEJ core complex proteins in response to IR in shCtrl vs. shUSP14-expressing H460 cells. As expected, chromatin recruitment of USP14 increased following IR treatment in a time-dependent manner in shCtrl cells (Figure 4A). However, in shUSP14 cells the levels of USP14 in chromatin fractions following IR was greatly reduced (Figure 4A). Consistent with the role of USP14 in regulating RNF168 in PCa cells, IR-induced chromatin-bound RNF168 was greatly increased in shUSP14- compared to shCtrl-expressing H460 cells (Figure 4A).

Additionally, in line with our previous studies in PCa cells, chromatin recruitment of NHEJ-associated factors, including Ku70, Ku80, Ligase IV, and XLF, was elevated in response to IR in shUSP14-expressing compared to shCtrl-expressing cells (Figure 4A). Next, a well-established host-cell reactivation reporter assay was used to examine whether USP14 regulates NHEJ DNA repair [26]. As shown in Figure 4B, shUSP14-expressing H460 cells had significantly increased levels of NHEJ compared to shCtrl- expressing cells, *p* < 0.001. 

A similar NHEJ-inhibitory role of USP14 was found in A549 cells using alternative markers of NHEJ. 53BP1 (Figure 4C,D, *p* < 0.001) and phosphorylation of DNA-PKcs on S2056 (Figure 4E,F, *p* < 0.001) IRIF formation, both of which are well established markers of NHEJ DDR signaling, were significantly higher in shUSP14- compared to shCtrl-expressing A549 cells. Overall, these data provide substantial evidence that USP14 inhibits NHEJ repair signaling in NSCLC cells.

### 2.5. Targeting USP14 Inhibits Homologous Recombination

Despite increased NHEJ, pharmacologic inhibition of USP14 using IU1 as well as USP14 knockdown resulted in increased sensitivity to IR. Since homologous recombination (HR) is an important determinant of the response to IR, we next investigated the role of USP14 in regulating HR. CtIP complex-dependent DNA end resection is an initial crucial step in HR repair as it generates a 3′ single-stranded DNA (ssDNA) [27]. The 3′ ssDNA tails are subsequently coated by RPA (replication protein A), which is composed of three subunits, RPA70, RPA32, and RPA14, which are often used as a marker for DNA end processing [28]. Rad51 then replaces RPA and forms extended filaments on the ssDNA [28]. These nucleoprotein filaments guide the ssDNA to pair with the homologous template DNA. Furthermore, BRCA1 is a HR-associated protein, which through its associations with multiple adaptor proteins, forms at least three distinct complexes (BRCA1-A, BRCA1-B, and BRCA1-C) in cells that critically regulate the HR pathway at multiple levels. Thus, BRCA1-A keeps HR under check by preventing excessive HR, and BRCA1-B and -C promote HR by increasing DNA end resection [29].

The accumulation of HR-associated DDR factors, i.e., RPA32 and BRCA1, was examined in response to IR in USP14-deficient NSCLC cells. shUSP14-expressing H460 cells had compromised accumulation of IRIF by RPA32 (Figure 5A,B, *p* < 0.0001) and BRCA1 (Figure 5D,E, *p* < 0.001). Poly (ADP ribose) polymerase (PARP) inhibition is known to kill HR deficient cells by synthetic lethality [30]. Indeed, a significantly reduced clonogenic survival in response to the PARP inhibitor rucaparib (Ruc) was observed in shUSP14- compared to shCtrl-expressing H460 cells (Figure 5C, *p* < 0.0001). Similarly, shUSP14-expressing A549 cells had significantly reduced IRIF formation by RPA32 compared to shCtrl-expressing cells (Figure 5F,G, *p* < 0.0001). Taken together, these data suggest that depletion of USP14 leads to HR deficiency in NSCLC cells.

Overall, these findings establish an important role for USP14 in regulating the cellular response to IR-induced DSBs. Depletion of USP14 results in an increase in NHEJ and deficiency in HR. An imbalance in the two major DSB repair pathways leads to unrepaired DSBs, and hence, renders the NSCLC cells more sensitive to IR-induced cell death.

## 3. Discussion

Here, we report for the first time that USP14: (i) has a role in regulating HR and (ii) is a target to increase radiosensitization in NSCLC. USP14 undergoes IRIF formation in NSCLC cells and its levels increase in chromatin-bound fractions in response to IR treatment. Upon further investigation of its role in DSB repair we found that, consistent with our previous findings in PCa, USP14 negatively regulates NHEJ in NSCLC. Importantly, USP14 promotes HR, thus targeting USP14 decreases the ability of NSCLC cells to repair DSBs induced by IR, leading to their radiosensitization.

DUBs play an important role in the regulation of DSB repair pathway choice between NHEJ vs. HR, and maintaining genomic stability [11]. Here, we have defined USP14 as a critical determinant of DSB repair pathway choice in NSCLC cells, and an important determinant of response to RT. USP14-proficient NSCLC cells have functional NHEJ and HR, and a balance between the two pathways accounts for limited radiosensitivity. However, USP14-deficient NSCLC cells have dysregulated DSB repair pathway choice, i.e., unchecked NHEJ and compromised HR, which leads to increased cell death in response to IR. In previous studies, several DUBs, including BRCC36 [31,32], USP1 [33], POH1 [34], and OTUB2 [34], have been shown to regulate pathway choice, and hence, the response to DNA damaging therapies.

While our study establishes an important role of USP14 in regulating both NHEJ and HR in NSCLC, the mechanism by which it does so is not clear. DUBs exert their effect by antagonizing the function of RNF8/RNF168: (i) BRCC36 [31] and POH1 [35] have been shown to remove K63 polyubiquitin chains at the DSB sites, (ii) USP3 [36], USP11 [37], USP26 [38], USP37 [38], and BAP1 [39] have been reported to deubiquitinate histones H2A/H2B, and (iii) OTUB2 has been shown to deubiquitinate L3MBTL1 [34]. In addition, we and others have shown that DUBs, including USP14 [21] and USP7 [40], regulate levels, while others, including OTUB1 [41], regulate the activity of RNF168. In previous studies we have shown that USP14 negatively regulates RNF168, an E3 Ub ligase that is essential for 53BP1 recruitment at DNA DSBs and activation of NHEJ [4,5,6,21]. Conversely, overabundance of RNF168 is known to promote mutagenic NHEJ repair, at the expense of HR [42]. Here, we show that USP14 knockdown increases chromatin recruitment of RNF168 and NHEJ core complex assembly, thus increasing NHEJ repair efficiency in NSCLC cells. Therefore, one possibility is that USP14 regulates RNF168 to keep NHEJ under check in order to prevent genomic imbalance resulting from mutagenic NHEJ that may be detrimental to the cells. Moreover, HR repair is compromised as a result of excessive NHEJ in USP14-depleted NSCLC cells. Consistently, we observe excessive recruitment of 53BP1 and NHEJ-core complex factors in NSCLC cells. Thus, RNF168-dependent up-regulation of NHEJ may underlie inhibition of HR. 

Another possibility is that USP14 may regulate the stability or recruitment of other downstream HR-associated factors. Deubiquitination-dependent regulation of NHEJ or HR repair-associated factors has been reported previously by our laboratory and others; thus, USP14 regulates the recruitment of Ku70 to DSB sites [22], USP4 regulates CtIP/MRE11 recruitment [43], and UCHL5 stabilizes NFRKB [44]. Thus, in future studies we will investigate whether USP14 regulates HR directly, through hyper-activation of NHEJ, or both.

Notably, we and others have found that USP14 mRNA and protein levels are up-regulated in NSCLC and that USP14 is a potential oncogene in NSCLC [17]. The Km plotter database suggests that high USP14 expression correlates with poor prognosis in NSCLC, and targeting USP14 has been shown to induce cell death in NSCLC [17]. In NSCLC and colon cancer cell lines, USP14 inhibition has been shown to induce cell death by targeting Wnt/β-catenin signaling pathway [17,45]. It may be noted that our data from clonogenic survival analyses indicate that targeting USP14 alone using IU1 led to varying degrees of cytotoxicities in NSCLC cell lines even in the absence of IR. This is expected since USP14 is a major DUB that modulates ubiquitination of a broad range of short-lived cellular proteins in the ubiquitin-proteasome system [15]. Additionally, as discussed above, inhibition of USP14 is likely to affect various additional biological processes besides DDR. However, the effect of USP14 inhibition on radiosensitization of NSCLC cell lines that we reveal stands out. Thus, there is significant decrease in clonogenic survival in most of the cell lines tested upon co-treatment with IU1 + IR, compared to IU1-alone, except for EBC1, which is a squamous cell line. Although limited data from one type of assay, one concentration of IU1, and one cell line may not be sufficient to come to a firm conclusion, nevertheless, our clonogenic and the TCGA data do suggest higher overexpression of USP14 in LUSC vs. LUAD. These findings may also translate into higher dependence, and hence, dramatic cell death with IU1, which may be investigated in more detail in future studies.

With the technical advancement for precise delivery, RT is becoming increasingly promising in the treatment of NSCLC. However, important questions that must be addressed include: (1) what molecular markers will identify the patients who will respond, and (2) what are the most effective combination treatment options based on the biomarkers that can distinguish responders from non-responders. USP14 prevents excessive NHEJ by regulating RNF168 and NHEJ core complex assembly and also is essential for proficient HR (Figure 6). Thus, we identify: (1) USP14 overexpression as a potential druggable target to increase radiosensitivity in NSCLC, and (2) a previously unexplored connection between USP14 and HR, known to regulate tumor cell response to RT. Overall this study contributes to mechanistic understanding of DDR in response to IR, which is instrumental to identifying rational drug combinations to enhance radiosensitization and improve treatment outcome in NSCLC patients.

## 4. Materials and Methods

### 4.1. Cell Culture and Treatments

A549, PC9, and EBC1 cells were a kind gift from Dr. George Stark (Lerner Research Institute, Cleveland Clinic, Cleveland, OH, USA). H460 cells were a kind gift from Dr. Mohammed Abazeed (Northwestern University, Evanston, IL, USA). 293T cells were from the American Type Culture Collection (ATCC, Rockville, MD, USA). Characteristics of the cell lines, including disease of origin and genetic alterations have been described in Table 1 [46]. 

Cells were maintained in RPMI medium containing l-glutamine, supplemented with 10% fetal bovine serum (Atlanta Biologicals, Lawrenceville, GA, USA), and 100× antibiotic-antimycotic (Gibco, Life Technologies, Gaithersburg, MD, USA). A549 and 293T cells were maintained in Dulbecco’s Modified Eagle’s Medium (DMEM). Cells were grown in a humidified incubator at 37 °C and 5% CO_2_. 

Cells were irradiated with doses, as indicated, at 25 °C, using a Mark I Irradiator (J. C. Shepherd & Associates, Irvine, CA, USA) with a 137Cs source emitting at a fixed dose-rate of 2.0 Gy/min, as described previously [15]. Cells were treated with 50 μM IU1, which is a noncovalent, specific inhibitor of USP14 and exhibits excellent selectivity for USP14 over the other DUBs [47], and 500 nM Nu7441 (Selleck Chemicals, Houston, TX, USA), wherever mentioned.

### 4.2. Plasmids

The following plasmids were used. Flag-HA-USP14 (22569) was purchased from Addgene (Watertown, MA, USA) [48]. ShRNA USP14 (TRCN0000007426), shControl (pLKO.1-puro), and lentiviral packaging plasmids, pCMV-VSV-G and pCMV-GAP-POL were from Sigma-Aldrich (St. Louis, MO, USA). 

### 4.3. Lentiviral Transduction of shRNA

293T cells were seeded in 10 cm plates. On the following day, they were transfected with shRNA plasmids (3 µg) together with the packaging plasmids (1 µg each) using the Fugene transfection reagent (Promega, Madison, WI, USA). At 24 h after transfection media was replaced with fresh growth media, and the media containing virus particles were collected. The following day, i.e., at 48 h post initial transfection, media containing viral particles was then pooled with the previous day’s media, and filtered through a 0.22-micron filter and subsequently used to infect H460 or A549 cells using polybrene (10 µg/mL). After 24 h incubation in a 37 °C incubator, the growth media was replenished and cells were subjected to puromycin selection (1 µg/mL) for 2 days, after which knockdown was validated using western blot.

### 4.4. Clonogenic Survival Assay

For clonogenic cell survival assay, 500–1000 cells were counted and plated in 6-well plates in triplicate. Following drug treatments, cells were allowed to grow for 14 days, fixed, and stained in methanol:acetic acid (75:25, *v*:*v*) containing 0.5% crystal violet (*w*:*v*) to visualize colonies of at least 50 cells.

### 4.5. Immunoblotting

Cells were lysed following the respective treatments in cell lysis buffer containing 50 mM Tris pH 7.4, 120 mM NaCl, 5 mM EDTA, 0.5% NP40, and 1 mM DTT, supplemented with the phosphatase inhibitor cocktail I, II (Sigma-Aldrich, St. Louis, MO, USA) and complete protease inhibitor tablet (Roche Diagnostics, Indianapolis, IN, USA). The sample was heated for 5 min in Laemmli buffer before separation by SDS–PAGE and immunoblotted with the indicated antibodies.

### 4.6. Confocal Immunostaining 

Confocal immunostaining was done as described before [22]. Briefly, 10^5^ cells/well were seeded on 22 × 22-mm coverslips in 6-well plates. Following the desired treatments, growth media was aspirated, and cells were washed with PBS, followed by 10 min fixing in 2.0% paraformaldehyde in phosphate buffered saline (PBS) at room temperature, followed by quenching in 0.1 M glycine solution in PBS. Then, the cells were permeabilized and blocked simultaneously with 0.1% Triton X-100/10% fetal bovine serum in PBS for 20 min. The coverslips were then immunostained using the primary antibodies diluted in blocking buffer, 0.1% Triton X-100 in PBS, overnight at 4 °C, followed by three washings in PBS. It was followed by staining with fluorescently-conjugated secondary antibodies for 1 h at room temp in dark, washed three times for 5 min each, and mounted using Vectashield containing DAPI. Images were collected using an HCX Plan Apo 40X/1.4N.A. oil immersion objective lens on a Leica TCS-SP2 confocal microscope (Leica Microsystems AG, Buffalo Grove, IL, USA). Quantification was based on data observed from at least 50 cells.

### 4.7. Chromatin Recruitment Assay

Chromatin extraction assay was used as described before [49] with minor adaptations. Thus, cells were harvested and washed with ice-cold phosphate-buffered saline. Cells were re-suspended in 80 µL of buffer A (10 mM Hepes, pH 7.9, 10 mM KCl, 1.5 mM MgCl_2_, 0.34 M sucrose, 10% glycerol, 0.1% Triton-X100, 1 mM dithiothreitol, phosphatase and protease inhibitors) and incubated for 15 min on ice, with gently tapping every 5 min, followed by centrifugation at low speed to separate cytoplasmic supernatant, from nuclear pellet fractions. Nuclear pellet was washed in 70 µL of buffer A, followed by low speed centrifugation. The supernatant was discarded, and the pellet was resuspended in 100 µL Buffer B (3 mM EDTA, 0.2 mM EGTA, 1 mM dithiothreitol, phosphatase, and protease inhibitors), and incubated for 30 min on ice, with gentle tapping every 4 min. The nucleoplasmic fraction supernatant, was then separated from the chromatin fraction pellet by slow speed centrifugation. Finally, the chromatin was re-suspended in 200 µL of 2× loading buffer and sheared by sonication.

### 4.8. Luciferase DNA Repair Reporter Assay

The luciferase assay for NHEJ was used as described previously [26].

### 4.9. Statistical Analyses

Statistical comparisons between two groups were conducted by using the Student’s *t*-test and between multiple groups using two-way ANOVA using the GraphPad Prism version 8.0.0 for Windows (GraphPad Software, San Diego, CA, USA).

## Figures and Tables

**Figure 1 ijms-21-06383-f001:**
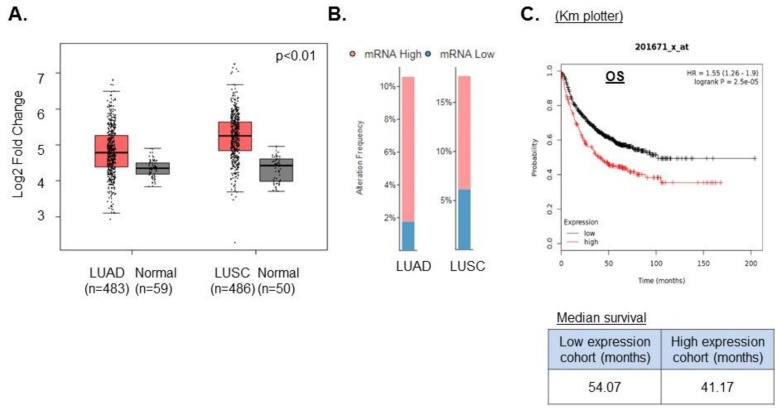
High USP14 expression correlates with poor prognosis in non-small cell lung cancer (NSCLC). (**A**) Differential USP14 mRNA expression analysis in lung adenocarcinoma (LUAD) vs. normal lung and lung squamous cell carcinoma (LUSC) compared to normal lung using the Gene Expression Profiling Interactive Analysis (GEPIA) database. (**B**) USP14 mRNA expression analysis in LUAD and LUSC using TCGA. (**C**) KM Plotter data for overall survival (OS) in low or high USP14-expressing LUAD patients.

**Figure 2 ijms-21-06383-f002:**
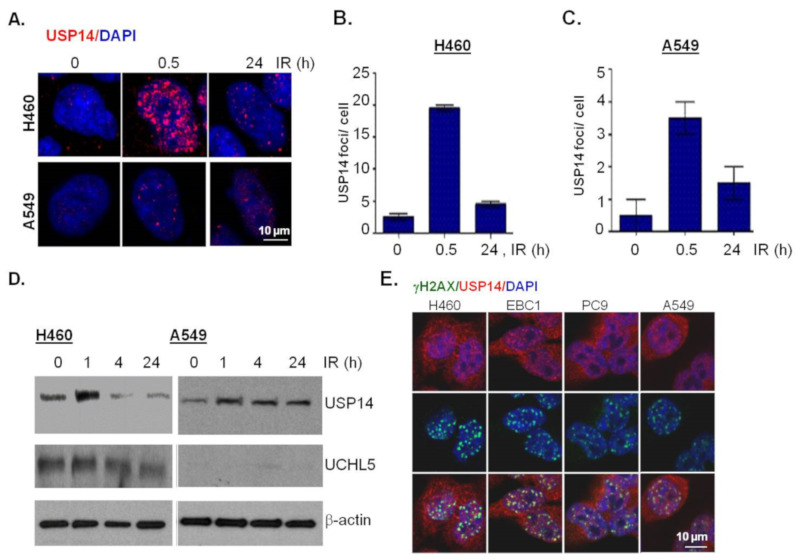
Ionizing radiation (IR) increases protein expression and double-strand break (DSB) recruitment of USP14. (**A**) Representative confocal images and (**B,C**) quantification of USP14 IR-induced foci (IRIF) at the indicated time points in H460 and A549 cells following 10 Gy IR treatment. Nuclei were stained with DAPI. (**D**) Western blot analysis for USP14 and UCHL5 in H460 and A549 cells at the indicated times after IR treatment. β-actin was used as the loading control. (**E**) Representative confocal images following co-immunostaining to show co-localization of USP14 and γH2AX IR-induced foci (IRIF) in the indicated cell lines at 1 h following 5 Gy IR treatment. Nuclei were stained with DAPI. Data shown are the means ± SEM (*n* = 3).

**Figure 3 ijms-21-06383-f003:**
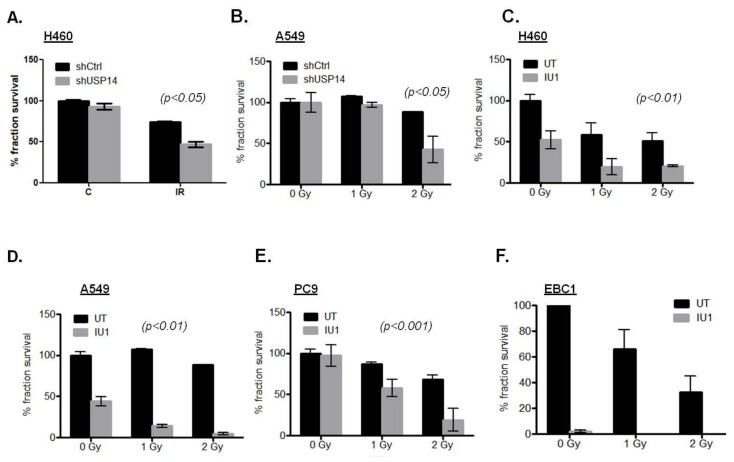
Targeting USP14 causes radiosensitization in NSCLC cells. Clonogenic cell survival analyses in shControl (shCtrl)- vs. shUSP14-expressing (**A**) H460 and (**B**) A549 cells treated with the indicated doses of IR. Clonogenic survival analyses in (**C**) H460, (**D**) A549, (**E**) PC9, and (**F**) EBC1 cells pretreated with 50 μM IU1 for 1 h +/− indicated doses of IR compared to those untreated (UT). Data shown are the means ± SEM (*n* = 2).

**Figure 4 ijms-21-06383-f004:**
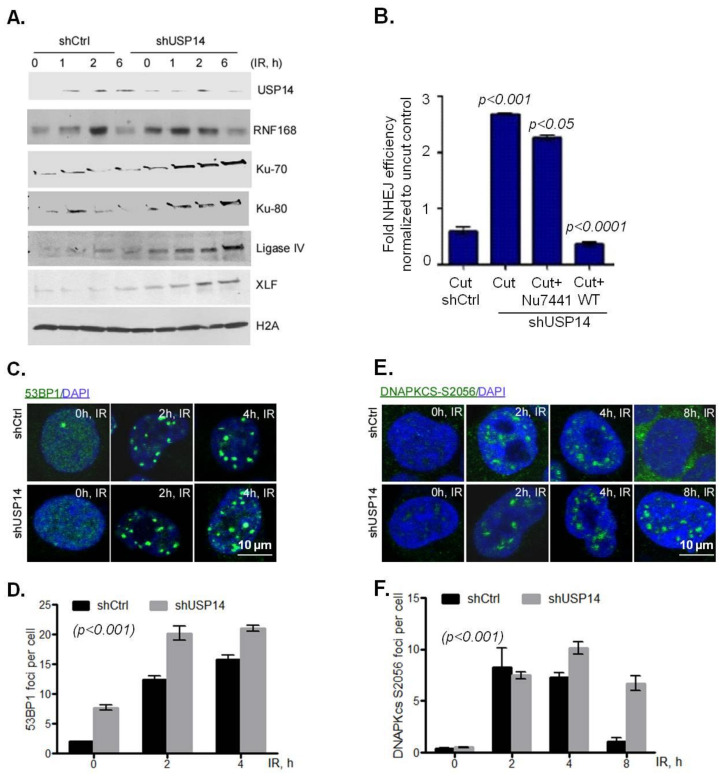
USP14 disrupts the non-homologous end joining (NHEJ) DNA damage response (DDR) in NSCLC cells. (**A**) Chromatin recruitment of the indicated proteins in H460 cells following IR +/− IU1 treatments at the indicated time points. H4 was used as loading control for the chromatin-bound fraction. (**B**) NHEJ measured by a plasmid luciferase repair assay. Data are normalized for transfection efficiency and then to uncut luciferase plasmid. Confocal immunostaining and graphical representation of quantitation of (**C**,**D**) 53BP1, and (**E**,**F**) DNA-PKCs S2056 at the indicated time points following 5Gy IR treatment in shControl (shCtrl)- vs. shUSP14-expressing A549 cells. Nuclei were stained with DAPI. Data shown are the means ± SEM (*n* = 2).

**Figure 5 ijms-21-06383-f005:**
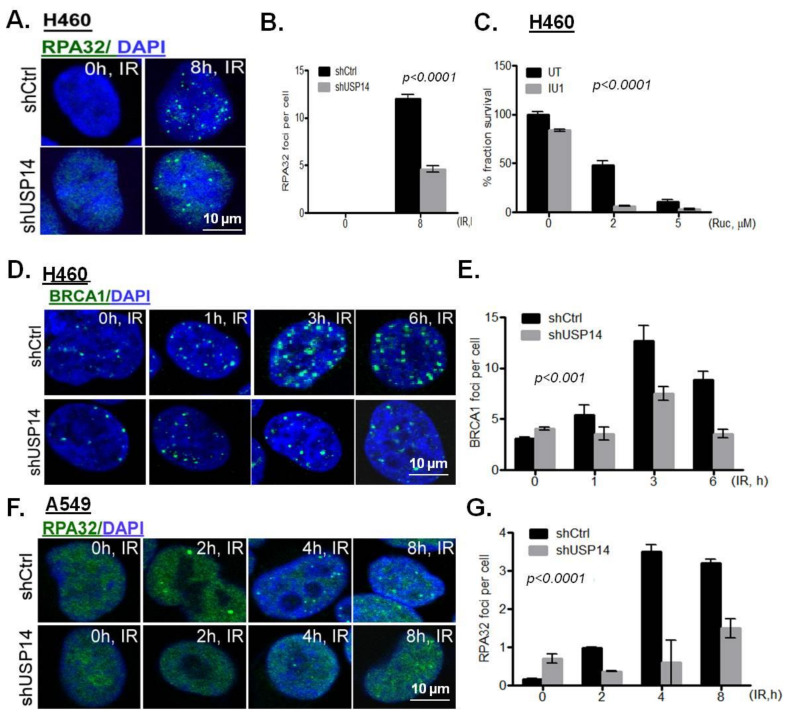
Targeting USP14 inhibits homologous recombination. (**A**,**B**) Confocal immunostaining and graphical representation of RPA32 IRIFs in shControl (shCtrl)- vs. shUSP14-expressing H460 cells at 8 h following 5 Gy IR treatment. Nuclei were stained with DAPI. (**C**) Clonogenic survival analysis in H460 cells pretreated with 50 μM IU1 for 1 h +/− indicated doses of rucaparib (Ruc). Confocal immunostaining and graphical representation of IRIFs for (**D**,**E**) BRCA1 in H460 and (**F**,**G**) RPA32 in A549 cells at the indicated time points in shControl (shCtrl)- vs. shUSP14-expressing cells. Nuclei were stained with DAPI. Data shown are the means ± SEM (*n* = 3).

**Figure 6 ijms-21-06383-f006:**
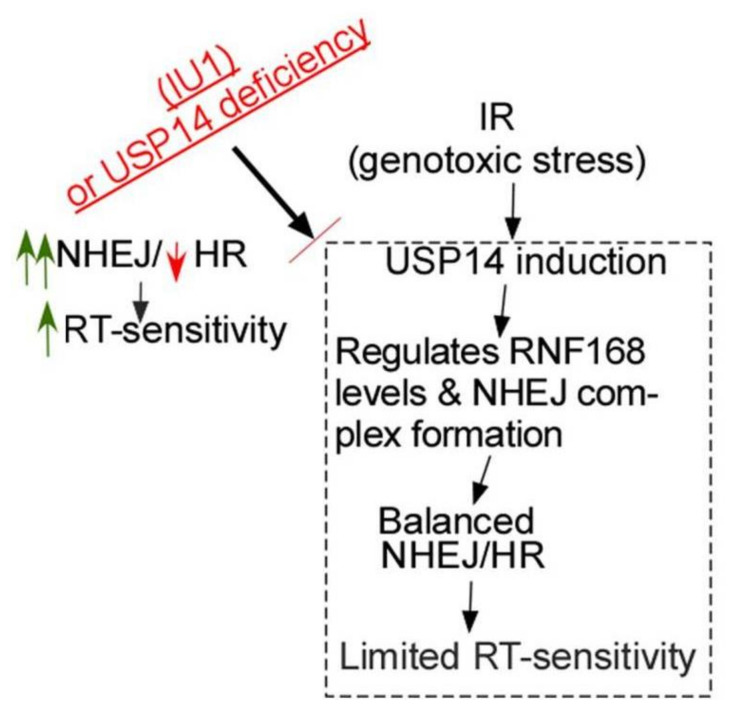
Model for the interface between USP14 and DDR in NSCLC. Genotoxic stress following IR treatment in NSCLC cells induces USP14 expression and its recruitment to DSB sites. USP14 regulates RNF168 and NHEJ core complex assembly to keep NHEJ under check and also maintains proficient HR. Optimal function of NHEJ and HR enables the cells to more efficiently repair DSBs induced by IR. Targeting USP14 causes up-regulation of NHEJ and inhibits HR. Excessive NHEJ is mutagenic, causing genomic instability and therefore, it is detrimental to the cells. Importantly, DSBs cannot be repaired efficiently in the absence of HR, thus, leading to increased cell death in response to IR. Those in red refer to inhibitory or downregulated, and in green refer to upregulated.

**Table 1 ijms-21-06383-t001:** Characteristics of cell lines used.

Cell Line	A549	H460	EBC1	PC9
Disease	Lung Adenocarcinoma	Lung large cell carcinoma	Lung squamous cell carcinoma	Lung Adenocarcinoma
Gene Alterations	KRAS: p.G12S; c.34G > A STK11: p.Q37X; c.109C > T CDKN2A: p.M1_*157del; c.1_471del47 SMARCA4: p.Q729fs; c.2184_2206del23	KRAS: p.Q61H; c.183A > T STK11: p.Q37X; c.109C > T MYCL: Amplification CDKN2A: p.ND; c.1_457del45 PI3CA: p.E545K; c.1633G > A	MET: Amplification TP53: p.E171X; c.511G > T	EGFR: p.E746_A750del; c.2235_2249del15 CDKN2A: p.G67V; c.200G > T TP53: p.R248Q; c.743G > A

ND: not defined.

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
