# Peer review of "USP14 Regulates DNA Damage Response and Is a Target for Radiosensitization in Non-Small Cell Lung Cancer"

_ijms, 2020, doi:10.3390/ijms21176383_

Round 1
Reviewer 1 Report
The presented research concerns an interesting issue that may improve response to radiation therapy in NSCLC in the future. The work presents promising results in which modern molecular tools were used to obtain them. The work, however, requires proofreading to improve the presentation of the results.
Line 6 - please add a country for affiliation 1 and 2
Introduction:
The introduction contains very little information on USP14, which is the main focus of the research presented. The proportions should be changed and a smaller portion should be devoted to the lack of response to RT and a larger portion to the USP14. There is also a lack of a precisely defined goal in the last paragraph of the introduction.
Results:
The presentation of the results should also be modified. Too many comments appear in the results. This section should only briefly present the discoveries made. However, comments should be moved to the discussion. An example is the reference to the results of a previous study in which the authors studied prostate cancer cells. In addition, some data, e.g. line 157-159, should be described in the introduction. What draws my attention is a nice presentation of the results in the figures.
Discussion:
The discussion does not seem to be extensive and contains few comparisons of the results of the authors' research with other studies.
Methods:
The methodology was presented correctly. Please add the company and country of the software used to the statistics GraphPad Prism - line 351
Author Response
Thank you for finding that our “research concerns an interesting issue that may improve response to radiation therapy in NSCLC” and that it “presents promising results in which modern molecular tools were used to obtain them”. Below we address the issues raised, indicating Lines (highlighted in yellow in the main text).
Line 6 - please add a country for affiliation 1 and 2
Country affiliation has now been added on lines 7 and 9
Introduction: The introduction contains very little information on USP14, which is the main focus of the research presented. The proportions should be changed and a smaller portion should be devoted to the lack of response to RT and a larger portion to the USP14. There is also a lack of a precisely defined goal in the last paragraph of the introduction.
Thank you for this suggestion. Following the Reviewer’s comment, we have included background on USP14 (lines 72-81), and clearly stated the goal in the last paragraph of introduction (lines 82-89).
Results: The presentation of the results should also be modified. Too many comments appear in the results. This section should only briefly present the discoveries made. However, comments should be moved to the discussion. An example is the reference to the results of a previous study in which the authors studied prostate cancer cells. In addition, some data, e.g. line 157-159, should be described in the introduction. What draws my attention is a nice presentation of the results in the figures.
Thank you for this suggestion. We have now revised the results section accordingly. We have deleted those comments in the Results section. Please see Lines 92, 104, 158.
Discussion: The discussion does not seem to be extensive and contains few comparisons of the results of the authors' research with other studies.
Thank you for this suggestion. We have now revised our discussion extensively, Lines 280, 287-295, 296-303, 307-308, 310-312, 313-318.
Methods: The methodology was presented correctly. Please add the company and country of the software used to the statistics GraphPad Prism - line 351
Thank you for this suggestion. We have now revised our methods accordingly, Lines 421-422.
Reviewer 2 Report
Despite advances in Non-small cell lung cancer (NSCLC) treatment, 5-year survival rate is still low. Radiotherapy response in patients being heterogeneous, identification of more effective treatment is crucial.
The authors identified USP14 as a regulator of double-strand break (DSB) repair pathways, in response to ionizing radiation (IR), in NSCLC. USP14 is a proteasomal deubiquitinase and IR increases levels and DSB recruitment of USP14 in NSCLC cell lines. USP14 knockdown or pharmacological inhibition increases radiosensitization in NSCLC cell lines. Moreover, shUSP14 increased Non-homologous end joining (NHEJ) efficiency, as indicated by chromatin recruitment of key NHEJ proteins and NHEJ reporter assay. Conversely, shUSP14 cells show decreased RPA32 and BRCA1 foci formation, suggesting Homologous Recombination (HR) -deficiency.
A broad comment is that the scientific approach is interesting. These results indicate that targeting USP14 is a promising strategy to improve cancer radiotherapy. However the analysis is well done, it suffers few caveats. Specific comments are listed hereafter:
Minor comments:
Page 2, lines 67-68. The introduction is excellent. Maybe some background on DSB repair inhibition (PARPi/in BRCA1 mutant context) would be helpful, although not in NSCLC context and not totally successful…
Page 2, line 87. Please correct “(…) of USP14 in NSCLC in normal lung vs. NSCLC.”
Page 2, line 91. Please correct “(…) We found that the expression of USP14 levels to be high in LUSC (10%) and lower in LUAD (8%) (Figure 1B).” Also invert LUAD and LUSC in Fig 1B to help readers.
Page 3, lines 111-118. Rogakou et al., princeps paper for gH2Ax could be cited. Please, give some background (here or in the Material & Methods part) for cells: mutations, origin, etc. as this is important for the work (for ex. but not necessary: Gene-Alteration Profile of Human Lung Cancer Cell Lines, Blanco et al.).
Page 10, line 259. Please correct “(…) may be 258 detrimental to the cell.s”
Page 12, line 323. Please correct “(…) The sample was heated for 5 min in Lamelli buffer…”
Page 12, line 348. The Luciferase assay is not described in the given reference.
Page 13, Abbreviations. Please double-check (typos in CtIP, H2AX = variant X of Histone 2A, typos in HR, PARP…)
Major comments:
Page 3, line 119. Please give more background on IU1 (in the Material & Methods part for ex). Did you use another inhibitor as control (for ex. P5091, selective inhibitor of USP7 or P22077)? In addition, 50 μM IU1 is well over the IC50. Please comment.
Page 5, lines 186-188. “(…) Depletion of USP14 results in an increase in NHEJ and deficiency in HR.” To my opinion, there is a higher recruitment of NHEJ factors… and potentially NHEJ repair, but the Luciferase assay could be the result of alternative repair mechanisms. Please comment. Also discuss the cell line choice, as different deficient cell lines would have been interesting in this context.
Page 11, lines 270-275. “…using IU1 led to varying degrees of cytotoxicities in NSCLC cell lines even in the absence of IR”. I noticed that USP14 immunofluorescence on A549 cells did not show a high level of foci (Fig2A) and, consistently, USP14 knockdown does not sensitize much these cells (Fig3B). However, in Fig3D, IU1 treatment highly sensitizes A549 cells. Please comment this point.
Author Response
Thank you for finding that our “scientific approach is interesting” and that “the analysis is well done”. Below we address the issues raised, indicating Lines (highlighted in yellow in the main text).
Minor comments:
Page 2, lines 67-68. The introduction is excellent. Maybe some background on DSB repair inhibition (PARPi/in BRCA1 mutant context) would be helpful, although not in NSCLC context and not totally successful…
Response: Relevant text Lines 55-57 and reference has been included as Ref. 9.
Remon J. et al., Somatic and Germline BRCA 1 and 2 Mutations in Advanced NSCLC From the SAFIR02-Lung Trial, JTO Clinical and Research Reports, 2020, 100068, ISSN 2666-3643, https://doi.org/10.1016/j.jtocrr.2020.100068.
Page 2, line 87. Please correct “(…) of USP14 in NSCLC in normal lung vs. NSCLC.”
We have now corrected as “We first searched the Gene Expression Profiling Interactive Analysis (GEPIA) database to identify the expression pattern of USP14 in normal lung vs. NSCLC,” Line 94.
Page 2, line 91. Please correct “(…) We found that the expression of USP14 levels to be high in LUSC (10%) and lower in LUAD (8%) (Figure 1B).” Also invert LUAD and LUSC in Fig 1B to help readers.
This correction has been made as “We found the expression of USP14 levels to be high in both LUAD (8%) and LUSC (10%) (Figure 1B).” Line 99
The LUAD vs LUSC data panels have been reversed to match the order in Figure 1A (please see Figure 1B).
Page 3, lines 111-118. Rogakou et al., princeps paper for gH2Ax could be cited.
Thank you for this suggestion, the following reference has been added, as Ref. 25. Please see Line 117
Rogakou, E. P., D. R. Pilch, et al. (1998). "DNA double-stranded breaks induce histone H2AX phosphorylation on serine 139." J Biol Chem 273(10): 5858-68.
Please, give some background (here or in the Material & Methods part) for cells: mutations, origin, etc. as this is important for the work (for ex. but not necessary: Gene-Alteration Profile of Human Lung Cancer Cell Lines, Blanco et al.).
Thank you very much for this comment, a Table has been included in the Material and Methods section describing the mutations and disease of origin for the various cell lines used in this study. Lines 352-355.
Page 10, line 259. Please correct “(…) may be 258 detrimental to the cell.s”
This inadvertent mistake has now been corrected, Line 309.
Page 12, line 323. Please correct “(…) The sample was heated for 5 min in Lamelli buffer…”
This inadvertent mistake has now been corrected, Line 388.
Page 12, line 348. The Luciferase assay is not described in the given reference.
We apologize for this mistake, the following correct reference has been added as Ref. 26
PTEN Regulates Nonhomologous End Joining By Epigenetic Induction of NHEJ1/XLF. Parker L. Sulkowski, Susan E. Scanlon, Sebastian Oeck and Peter M. Glazer.DOI: 10.1158/1541-7786.MCR-17-0581 Published August 2018
Page 13, Abbreviations. Please double-check (typos in CtIP, H2AX = variant X of Histone 2A, typos in HR, PARP…)
We have carefully checked the text for any typos.
Major comments:
Page 3, line 119. Please give more background on IU1 (in the Material & Methods part for ex). Did you use another inhibitor as control (for ex. P5091, selective inhibitor of USP7 or P22077)? In addition, 50 μM IU1 is well over the IC50. Please comment.
Response: This is a valid question. We agree with the Reviewer that our data from clonogenic survival analyses indeed indicate that targeting USP14 alone using IU1 led to varying degrees of cytotoxicities in NSCLC cell lines even in the absence of IR.
UPS14 is a major DUB that modulates ubiquitination of a broad range of short-lived cellular proteins in the ubiquitin-proteasome system. In various cancers, USP14 inhibition has been shown to induce cell death by targeting the Wnt/β-catenin and TGFβ/Smad pathways. Thus, inhibition of USP14 is likely to affect various additional biological processes besides DDR. Nevertheless, the effect of USP14 inhibition in inducing radiosensitization of NSCLC cell lines we have found stands out, there is significant decrease in clonogenic survival in most of the cell lines tested upon co-treatment with IU1+IR, compared to IU1-alone, except for EBC1 which is a squamous cell line. Although data from one type of assay, one concentration of IU1, and one cell line may not be sufficient to conclude but our clonogenic and TCGA data do suggest higher overexpression of USP14 in LUSC vs. LUAD which may also translate into higher dependence and hence dramatic cell death with IU1, which may be investigated in further details in the future studies. This aspect has been included in the discussion. Following reviewer’s comment discussion has been further modified . Lines 329-335.
Moreover, stable expression of shUSP14 in NSCLC cell lines led to significant reduction in clonogenic survival in response to IR. Thus, our data with both genetic and pharmacological targeting of USP14 provide proof of principle supporting USP14 being a potential target to increase radiosensitization.
We have chosen a 50 µM concentration based on literature, as indicated below:
- B.H. Lee et al. Enhancement of proteasome activity by a small-molecule inhibitor of USP14. Nature 467, 179–184 (2010). “When added at 50 μM, IU1 reached an apparent intracellular concentration of ~13 μM within 1 hour, ’ To determine whether IU1 could enhance proteasome function in cells, we expressed Tau in MEFs treated with sub-cytotoxic doses of IU1. IU1 induced dose-dependent reduction in Tau levels, with a strong effect seen at 50 μM (Fig. 4a; Supplementary Fig. 23)”…
- Daichao Xu, et al. USP14 regulates autophagy by suppressing K63 ubiquitination of Beclin 1. Genes & Dev. 2016. 30: 1718-1730.
- Xiaohong Xia, et al. Inhibition of USP14 enhances the sensitivity of breast cancer to enzalutamide, Journal of Experimental & Clinical Cancer Research volume 38, Article number: 220 (2019)
Also, we have now included background on IU1 in the Materials & Methods section. Lines 362-363.
Page 5, lines 186-188. “(…) Depletion of USP14 results in an increase in NHEJ and deficiency in HR.” To my opinion, there is a higher recruitment of NHEJ factors… and potentially NHEJ repair, but the Luciferase assay could be the result of alternative repair mechanisms. Please comment. Also discuss the cell line choice, as different deficient cell lines would have been interesting in this context.
Response: We sincerely apologize for the confusion, the reporter assay used is indeed an established method to study NHEJ efficiency, as described in the following reference. A correction has been made in the main text as well Ref. 26.
Parker L Sulkowski, at al. PTEN Regulates Nonhomologous End Joining by Epigenetic Induction of NHEJ1/XLF, Mol Cancer Res. 2018 Aug;16(8):1241-1254.
Regarding the cell line, the H460 cell line was chosen for this assay as a standard cell line used in radiation response studies in lung cancers since it is radiosensitive. For our experiments we choose this cell line also because it has high expression of USP14, so it was suitable for genetic knockdown and overexpression studies.
Page 11, lines 270-275. “…using IU1 led to varying degrees of cytotoxicities in NSCLC cell lines even in the absence of IR”. I noticed that USP14 immunofluorescence on A549 cells did not show a high level of foci (Fig. 2A) and, consistently, USP14 knockdown does not sensitize much these cells (Fig. 3B). However, in Fig. 3D, IU1 treatment highly sensitizes A549 cells. Please comment this point.
Response: Thank you for this comment. Our data suggest an important role for USP14 in regulating DDR in NSCLC, both NHEJ and HR. This conclusion is based on multiple observations, as shown in the manuscript and listed below, points 3A- 3E. In addition, we also indicate in the Discussion section that our data and other studies suggest that regulation of DDR may not be the only effect observed upon targeting USP14, since USP14 has been shown to target additional pro-survival pathways. The discrepancy observed between IU1 treatment vs. USP14 knockdown may also be the result of an incomplete depletion of USP14 by shRNA-mediated knockdown. Thus, residual USP14 may be sufficient to keep NHEJ under check and at the same time allow functional HR to repair IR-induced DSBs, and thus induce lower clonogenic inhibition. In future studies, this may be tested through CRISPR-Cas9 knockout of USP14 as well as by doing detailed studies of dose response to IU1+IR.
3A. USP14 foci formation in response to IR (Fig. 1A &B).
3B. IR-induced USP14 nuclear foci co-localize with γH2AX foci (which is the gold standard marker for DBSs), Fig.1D.
3C. Data from clonogenic survival assays suggests maximum and significant reduction in clonogenic growth in NSCLC cell lines treated with IR following genetic or pharmacological targeting of USP14 compared to IR alone or targeting USP14 alone (Fig. 3).
3D. Increased recruitment of NHEJ factors in shUSP14 expressing NSCLC cell lines, as shown by chromatin recruitment analysis, NHEJ reporter analysis, and DNAPKcsT2609 and 53BP1 foci formation in response to IR (Fig. 4).
3E. Decreased recruitment of HR-associated factors, including BRCA1 and RPA32 in NSCLC cell lines (Fig. 5).
Round 2
Reviewer 2 Report
Dear Authors,
Thank you for the corrections,
I wish you good luck with your future work.